

# Weighted multiple testing procedures in genome-wide association studies

Ludivine Obry and Cyril Dalmasso

Université Paris-Saclay, CNRS, Univ Evry, Laboratoire de Mathématiques et Modélisation d'Evry, Evry-Courcouronnes, France

## ABSTRACT

Multiple testing procedures controlling the false discovery rate (FDR) are increasingly used in the context of genome wide association studies (GWAS), and weighted multiple testing procedures that incorporate covariate information are efficient to improve the power to detect associations. In this work, we evaluate some recent weighted multiple testing procedures in the specific context of GWAS through a simulation study. We also present a new efficient procedure called wBHa that prioritizes the detection of genetic variants with low minor allele frequencies while maximizing the overall detection power. The results indicate good performance of our procedure compared to other weighted multiple testing procedures. In particular, in all simulated settings, wBHa tends to outperform other procedures in detecting rare variants while maintaining good overall power. The use of the different procedures is illustrated with a real dataset.

Corresponding author
Ludivine Obry,
ludivine.obry@univ-evry.fr

## INTRODUCTION

### Multiple testing procedures in genome-wide association studies

In genome-wide association studies (GWAS), hundreds of thousands of genetic markers, (usually single nucleotide polymorphisms—SNPs), are simultaneously tested for an association with a phenotype of interest. In this context, the most common approach consists in using single-marker methods (*Bush & Moore, 2012*). To avoid an increase in false significant results, multiple testing procedures are then applied with the objective to control a global error rate.

The two major multiple testing error criteria used in GWAS are the family wide error-rate (FWER), which is defined as the probability of obtaining at least one false positive, and the false discovery rate (FDR), which is defined as the expected proportion of falsely rejected hypotheses over all rejected hypotheses (*Benjamini & Hochberg, 1995*). The latter criterion was designed as an error criterion less stringent than the FWER. The authors also included an FDR controlling procedure (hereafter referred to as BH).

In GWAS, the FWER has been traditionally used to control multiplicity with methods such as the classical Bonferroni procedure. To account for the correlation structure induced by linkage disequilibrium between SNPs, different approaches have been proposed to derive significant thresholds based on an estimation of the effective number of independent SNPs (*Pe'er et al., 2008*; *Dudbridge & Gusnanto, 2008*; *Gao, Starmer & Martin,*

*2008*; *Duggal et al., 2008*; *Galwey, 2009*; *Li et al., 2012*; *Xu et al., 2014*). However, multiple testing strategies based on controlling the FWER are known to be overly conservative when the number of tests is large. Thus, the FDR has become increasingly popular in the context of GWAS, where obtaining a few false positives can be considered as acceptable (*Brzyski et al., 2017*). While correlations between SNPs can substantially deteriorate the performance of many FDR procedures (*Owen, 2005*; *Qiu, Klebanov & Yakovlev, 2005*; *Sarkar, 2006*; *Efron, 2007*; *Neuvial, 2008*), the classical FDR procedures remain valid under different dependence assumptions (*Benjamini & Yekutieli, 2001*; *Farcomeni, 2007*; *Wu et al., 2009*). In particular, *Sabatti, Service & Freimer (2003)* observed that the validity holds for the classical BH procedure in case–control studies. Thus, in a GWAS context with correlated tests, FDR-based procedures achieved higher power than the FWER-based strategy, even at a strict FDR level (*Otani et al., 2018*).

This single-marker strategy combined with multiple testing procedures has made it possible to identify hundreds of genetic variants associated with several diseases, usually close to previously unsuspected genes. However, only a small fraction of the phenotypic variations has been explained, reflecting a large part of missing heritability (*Maher, 2008*; *Manolio et al., 2009*; *Tam et al., 2019*). Many reasons for this missing heritability have been proposed, such as common variants with small effects which have yet to be discovered, the difficult identification of dominance genetic variation and epistasis, and rare variants with strong effects that are poorly detected by genotyping arrays (*Eichler et al., 2010*; *Zuk et al., 2014*).

Indeed, rare variants are poorly covered by genotyping arrays and are usually filtered from GWAS, since the overall idea is that frequent genetic variants explain a large part of the heritability in common diseases and are easier to detect in populations (*Panagiotou, Evangelou & Ioannidis, 2010*; *Riancho, 2012*; *Korte & Farlow, 2013*). Thus, markers with a minor allele frequency (MAF) lower than a specific threshold (usually 1% or 5%) are removed from the analyses in order to limit problems due to too small sample sizes. However, several studies have recently shown that the initial assumption of GWAS is relatively false, so the full potential of these studies has not been unlocked since a part of the missing heritability can be partially explained by rare variants that are difficult to detect (*Manolio et al., 2009*; *Auer & Lettre, 2015*; *Bandyopadhyay, Chanda & Wang, 2017*). In fact, these variants are likely to have greater effects size than common variants (*Janssens et al., 2007*; *Bodmer & Bonilla, 2008*; *Marouli et al., 2017*).

Various approaches have been proposed to address the problem of detecting rare variants. A popular strategy consists in using burden tests that collapse rare variants in a genomic region into a single burden statistic (*Morgenthaler & Thilly, 2007*; *Li & Leal, 2008*; *Madsen & Browning, 2009*; *Wu et al., 2011*; *Lee et al., 2014*; *Li et al., 2019*). However, with this strategy, only the cumulative effect of the SNPs contained in each set is tested. To keep information on individual markers, it has been shown that considering weighting strategies as described below is an effective way to increase the detection power of rare variants with high genetic effects (*Dalmasso, Génin & Trégouet, 2008*).

## Weighting strategies

One of the drawbacks of standard multiple testing approaches is that all hypotheses (that correspond to SNPs in the context of GWAS) are considered as interchangeable. However, the statistical or biological properties of individual tests are usually different, so some tests have greater power than others. Moreover, classical multiple testing methods do not use prior knowledge which can improve the detection power of associated variants (*Roeder & Wasserman, 2009*; *Gui, Tosteson & Borsuk, 2012*). Using weights is a way to increase this detection power while maintaining the error rate level.

The principle of weighted multiple testing procedures is to multiply the thresholds by weights (or equivalently the *p*-values or the test statistics by inverse weights) (*Holm, 1979*; *Benjamini & Hochberg, 1997*; *Genovese, Roeder & Wasserman, 2006*). Thus, the power increases for some individual hypotheses and it decreases for others, while keeping the error criterion control at an average weight equal to 1. In practice, most weighting procedures deal with a list of weighted pvalues. Recently, several procedures controlling the FDR have been proposed (*Genovese, Roeder & Wasserman, 2006*; *Scott et al., 2015*; *Ignatiadis et al., 2016*; *Lei & Fithian, 2018*; *Boca & Leek, 2018*; *Li & Barber, 2019*; *Zhang, Xia & Zou, 2019*; *Zhang & Chen, 2020*).

We distinguish two main approaches defining weights. The first consists in defining external weights, based on prior scientific knowledge of the data (*Genovese, Roeder & Wasserman, 2006*; *Roeder et al., 2006*; *Hu, Zhao & Zhou, 2010*). The second relies on adaptive procedures that focus on optimal weights estimated from the data (*Wasserman & Roeder, 2006*; *Roeder, Devlin & Wasserman, 2007*; *Roquain & Wiel, 2008*; *Roeder & Wasserman, 2009*; *Zhao & Zhang, 2014*; *Zhao & Fung, 2016*; *Durand, 2019*). In the latter approach, different methods using informative covariates for maximizing the overall power have recently been introduced (*Ignatiadis et al., 2016*; *Zhang & Chen, 2020*). In a GWAS context, using the MAF as an informative covariate can help to detect rare variants.

## Objectives

In this work, we evaluate recent weighted FDR controlling procedures in the specific context of GWAS. We also introduce a new adaptive procedure called wBHa in order to prioritize the detection of genetic markers having a low MAF by letting the procedure adapt a weighting function in order to maximize the overall power.

To evaluate the procedures, we conducted an extensive simulation study. Among the procedures using informative covariates to define weights, we considered wBH (*Genovese, Roeder & Wasserman, 2006*), FDRreg (*Scott et al., 2015*), IHW (*Ignatiadis et al., 2016*), swfdr (*Boca & Leek, 2018*), AdaPT (*Lei & Fithian, 2018*), SABHA (*Li & Barber, 2019*), AdaFDR (*Zhang, Xia & Zou, 2019*), and CAMT (*Zhang & Chen, 2020*). We also included two unweighted procedures: BH (*Benjamini & Hochberg, 1995*) and qvalue (*Storey & Tibshirani, 2003*).

In the next section, we briefly describe the statistical framework, the main evaluated methods, and present our new wBHa procedure. We also detail the simulation study that was conducted to evaluate the procedures. The following section presents the results of the

**Table 1  Outcomes for *m* tested hypotheses in a multiple testing situation.**

|  | $H_0$ not rejected | $H_0$ rejected | Total |
|---|---|---|---|
| True $H_0$ | TN | FP | $m_0$ |
| False $H_0$ (True $H_1$) | FN | TP | $m_1$ |
|  | $W = m - R$ | $R$ | $m$ |

simulation study and we illustrate the use of our method with a real public dataset. Finally, in the last section, we conclude with a discussion on the different approaches.

## METHODS

### Statistical setting

Let $m$ denote the total number of null hypotheses tested. Among them, $m_0$ null hypotheses ($H_0$) are true while $m_1$ null hypotheses are false, *i.e.*, $m_1$ alternative hypotheses ($H_1$) are true. When a multiple testing procedure is applied, $R$ null hypotheses are rejected and $W = m - R$ null hypotheses are not rejected. The different outcomes are summarized in Table 1. Only the random variable $R$ can be observed while the numbers of true positives ($TP$), false positives ($FP$), true negatives ($TN$) and false negatives ($FN$) are unobservable random variables.

Using these notations, the family-wise error rate is defined as $FWER = \mathbb{P}(FP > 0)$ while the false discovery rate is defined as $FDR = \mathbb{E}\left(\frac{FP}{R \vee 1}\right)$. In the following, we focus on $FDR$ controlling procedures, most of them being based on a preliminary calculation of $p$-values.

Let $p_i\ i = (1, \ldots, m)$ be the $p$-values calculated for the $m$ tested hypotheses and let $P$ be the corresponding random variables. In the classical two components mixture model, the population of $p$-values can be viewed as a mixture of two sub-populations corresponding to null and alternative hypotheses. Thus, the marginal distribution of each $p$-value can be written:

$$f(p) = \pi_0 f_0(p) + (1 - \pi_0) f_1(p) \tag{1}$$

where $f_0$ denotes the null density, $f_1$ the alternative density, and $\pi_0 = Pr(H_i = 0)$ and $\pi_1 = Pr(H_i = 1)$ with $H_i$ being the random variable such that $H_i = 0$ if the null hypothesis is true, $H_i = 1$ if the alternative hypothesis is true. Note that if the test statistics are continuous random variables, then, under the null hypothesis, the $p$-values follow a uniform distribution over the interval $[0, 1]$.

Finally, let $X = (x_i, \ldots, x_m)$ be an informative covariate defined as any continuous or categorical variable, independent from the $p$-values under the null hypothesis, and informative on the statistical properties of the hypothesis tests. In this work, we consider MAF as covariate to prioritize the detection of rare SNPs with strong effects over more frequent SNPs with weaker effects, since the individual power is lower for SNPs with low MAF than for common variants. Nevertheless, other covariates can be considered, such as the signal quality, sample size, or the distance between the genetic variant and the genomic location of the phenotype in expression-QTL analysis, *etc.* (*Ignatiadis et al., 2016*; *Korthauer et al., 2019*).

We briefly describe hereafter the different weighted multiple testing procedures included in our evaluation.

## Unweighted multiple testing procedures
### Benjamini and Hochberg

The linear step-up procedure (BH) proposed by *Benjamini & Hochberg (1995)* to control the FDR at level $\alpha$ consists in rejecting all $k$ null hypotheses corresponding to the $k$ smallest $p$-values where $k = max(i \geq 0 : p_{(i)} \leq \frac{i\alpha}{m})$, $p_{(i)}$ being the ordered $p$-values. It has been shown that when the test statistics are PRDS (positive regression dependent on subset of null statistics), the BH procedure controls the FDR at level $\pi_0\alpha$ (*i.e.*, $FDR \leq \pi_0\alpha$) (*Benjamini & Hochberg, 1995*; *Benjamini & Yekutieli, 2001*).

### Qvalue

To improve the power of the BH procedure, several adaptive procedures have been introduced in which the proportion of true null hypotheses $\pi_0$ is estimated from the data (*Storey, 2002*; *Dalmasso, Broët & Moreau, 2005*; *Benjamini, Krieger & Yekutieli, 2006*; *Liang & Nettleton, 2012*). One of the most used is the Qvalue procedure, in which a cubic spline based method is used to estimate the quantity $lim_{\lambda \to 1}\hat{\pi}_0(\lambda)$ where $\hat{\pi}_0(\lambda) = \frac{\#\{p_i > \lambda\}}{m(1-\lambda)}$ (*Storey, 2002*; *Storey & Tibshirani, 2003*).

## Weighted multiple testing procedures
### Weighted Benjamini and Hochberg

The weighted BH procedure (wBH) was introduced by *Genovese, Roeder & Wasserman (2006)*. It consists in assigning to each null hypothesis $H_{0,i}$ a non-negative weight such that $\sum_{i=1}^{m} w_i = m$. Then, the BH procedure is applied by replacing $p_i$ by $\frac{p_i}{w_i}$. This procedure has been proven to control the FDR.

### False discovery rate regression

The false discovery rate regression (FDRreg) procedure introduced by *Scott et al. (2015)* is an adaptive procedure in which the proportion of true null hypotheses $\pi_0$ is estimated. However, this quantity is made dependent on the covariate, leading to: $\pi_0(x_i) = Pr(H_i = 0|X_i = x_i)$ and $FDR(x_i) = \mathbb{E}(\frac{FP}{max(R,1)}|X_i = x_i)$ with $\pi_0(x_i)$ representing weights specific to each hypothesis. Thus, noting $z_i$ the test statistics, the two-components mixture model can be written:

$$f(z_i) = \pi_0(x_i)f_0(z_i) + (1 - \pi_0(x_i))f_1(z_i) \tag{2}$$

In this approach, the alternative density $f_1(z_i)$ is taken to be a location mixture of null density which is assumed to be Gaussian. The mixing proportion of $f_1(z_i)$ is fitted *via* a predictive recursion algorithm (*Newton, 2002*) and the model parameters in Eq. (2) are then estimated by an EM algorithm treating the mixing proportions of the alternative distribution as fixed. A fully Bayesian approach based on MCMC method is also proposed.

### Science-wise false discovery rate

The science-wise false discovery rate (swfdr) procedure introduced by *Boca & Leek (2018)* is similar to FDRreg in the sense that $\pi_0$ and $FDR$ are extended by conditioning on the

covariate. However, while FDRreg jointly estimates $\pi_0$ and the *FDR* by assuming that the test statistics are normally distributed, swfdr first estimates the proportion of true null hypotheses, and then the *FDR* is obtained from a plug-in estimator. To estimate the proportions $\pi_0(x_i)$, an approach similar to Qvalue is proposed but the ratio $\frac{\#\{p_i > \lambda\}}{m(1-\lambda)}$ is replaced by $\frac{\hat{\mathbb{E}}(1_{P_i > \lambda} | X_i = x_i)}{(1-\lambda)}$ where $\mathbb{E}(1_{P_i > \lambda} | X_i = x_i)$ is estimated from a logistic regression model.

### Covariate adaptive multiple testing

The covariate adaptive multiple testing (CAMT) procedure introduced by *Zhang & Chen (2020)* is also based on the mixture model (Eq. (2)) with mixing proportions dependent on the covariate. However, this procedure relies on the local version of the FDR, the local false discovery rate (*lfdr*), introduced by *Efron et al. (2001)*. The lfdr is defined as the posterior probability that a hypothesis is null given a specific pvalue: $lfdr(p_i) = Pr(H = 0 | P = p_i) = \frac{\pi_0 f_0(p_i)}{f(p_i)}$.

From this definition, the FDR can be derived by the relationship $FDR = \mathbb{E}(lfdr | P \in \Gamma)$ where $\Gamma$ is a rejection region for the pvalues (*Efron et al., 2001*; *Dalmasso, Bar-Hen & Broët, 2007*). In addition, the optimal decision rule can be written $lfdr(p_i) \le t \Leftrightarrow \frac{f_{1,i}(p_i)}{f_0(p_i)} \ge \frac{(1-t)\pi_0(x_i)}{t(1-\pi_0(x_i))}$.

The principle of CAMT is to replace the ratio $\frac{f_{1i}}{f_0}$ in the optimal decision rule by a surrogate function $h_i(p) = (1 - k_i) p_i^{-k_i}$. Then, the parameters $\pi_0(x_i)$ and $k_i$ are estimated from an EM algorithm in order to find the optimal threshold $t$ that allows the FDR to be controlled at the desired level.

### Independent Hypothesis Weighting

The Independent Hypothesis Weighting procedure (IHW) was introduced by *Ignatiadis et al. (2016)*. Here, the objective is to find optimal weights that maximize overall power. The basic idea is to divide hypotheses into $G$ groups according to the ordered values of the covariate. Then, positive weights are assigned to each group $g$ in order to maximize the number of rejections.

To avoid overfitting, the authors introduced a hypothesis splitting approach which consists in randomly splitting the $m$ hypotheses into $k$ folds independently of the pvalues and covariates. For each fold, an optimization problem is applied to the hypotheses of the $k - 1$ remaining folds in order to derive weights $\tilde{\omega}_g$ ($g = 1, \ldots, G$) that maximize the overall power. Then, hypotheses of the held out fold lying in the group g are assigned weight $\tilde{\omega}_g$.

To make the optimization problem convex, the authors proposed using the Grenander estimator instead of the empirical cumulative distribution function. In addition, to solve the optimization problem, they added a regularization parameter $\lambda$ such that $\sum_{g=2}^{G} \| w_g - w_{g-1} \| \le \lambda$ where $\lambda > 0$. This regularization parameter allows for weights of successive groups to be relatively similar.

Finally, once the weights are estimated, a standard wBH procedure is applied with the resulting weights vector.

## wBHa procedure

In a context of GWAS where rare variants have strong genetic effects, it has been shown that using a weighted-Holm procedure with weights depending on MAF can substantially improve the power to detect associations (*Dalmasso, Génin & Trégouet, 2008*). Here, we extend this approach to the control of the FDR in order to prioritize the detection of rare variants while optimizing the overall detection power. The principle of our method, called wBHa, is to define weights as an explicit function of the covariate by:

$$w(x_i, a) = \frac{m}{\sum_{j=1}^{m} \frac{1}{x_j^a}} \times \frac{1}{x_i^a} \tag{3}$$

In the following, we set $x_i = MAF_i$ to prioritize the detection of rare variants, but as mentioned before, other covariates can be considered, the proposed function being particularly adapted to any continuous informative covariate. Thus, our procedure is similar to the classical weighted BH procedure with weights prioritizing the detection of variants with low MAF. However, the introduction of the free parameter $a$ makes it possible to optimize weights flexibly in order to improve the overall power.

In practice, the naive algorithm to obtain the optimal $a$ consists in choosing the value leading to the maximum number of rejection $R$ for a grid of values. In the event of tied values (*i.e.,* if different $a$ values lead to the same number of rejections), the optimal $a$ is set to the largest value of the longest interval defined by consecutive $a$ values leading to the largest number of rejections. The steps leading to the optimal $a$ are presented in Algorithm 1. Once the optimal $a$ is obtained, wBH procedure is applied with the corresponding weights.

To avoid overfitting, we consider a bagging approach (*Breiman, 1996*; *González et al., 2020*): $K$ datasets are generated by sampling $\frac{m}{K}$ $p$-values with replacement within the $m$ tested hypotheses. For each dataset, the naive algorithm is applied leading to a set of $K$ values: $a_1, \ldots, a_K$. Finally, the optimal $a$ value is obtained by calculating the average value of all $a_k$ values.

By choosing $\frac{m}{K}$ for bagging sample sizes, it becomes possible to speed up the algorithm and to increase diversity for further differentiation of the samples of the hypothesis set. In addition, while standard bagging generates bootstrap samples of equal size as the original dataset, optimal results are often obtained with sampling ratios smaller than the standard choice (*Martínez-Muñoz & Suárez, 2010*).

To sum up, our wBHa procedure not only enables the incorporation of external information to improve the detection power of rare variants but also makes it possible to search for optimal weights by maximizing the overall power for a class of weights defined as a function of a free parameter $a$.

In the next section, we describe the simulation study we performed to evaluate the different procedures in a context of GWAS.

---

**Algorithm 1:** *a* Optimization Algorithm

---

**Input:** A $m$-tuple of $p$-values $P = (p_1, \ldots, p_m)$ and covariates $X = (x_i, \ldots, x_m)$, a nominal level $\alpha \in (0, 1)$ for the FDR and a number of folds $K = 100$.

**Output:** Optimal $a$

**for** $k_i = 1, \ldots, K$ **do**

   Sampling with remplacement of $\frac{m}{K}$ hypotheses;

   **for** $a = 0, 0.1, 0.2, \ldots, 10$ **do**

      Application of wBH procedure at level $\alpha$ with $w(x_i, a) = \frac{m}{\sum_{j=1}^{m} \frac{1}{x_j^a}} \times \frac{1}{x_i^a}$;

      Computation and saving of the numbers of rejections $R$;

   **end**

   Saving the values $a$ leading to the maximum of $R$ in an ordered $L$-tuple ($L \geq 1$) $A = (a_1, \ldots, a_L)$;

   **if** $L > 1$ **then**

      Computation of the successive differences in $A$ ;

      Definition of interval bounds from differences larger to the step 0.1 ;

      Clustering of the $L$ values of $A$ within the $v$ intervals thus defined ;

      **if** $v = 1$ **then**

         Saving the maximum value of the vector $A$ ;

      **else**

         Computation of the length of each interval ;

         **if** *one of the intervals is longer than the others* **then**

            Saving the maximum value in the longest interval ;

         **else**

            Saving the maximum value in the interval closest to 1 ;

         **end**

      **end**

   **end**

**end**

Optimal $a$ obtained by calculating the average of the $K$ values ;

---

# SIMULATION STUDY

To compare and evaluate the performance of the different procedures in a GWAS context, we performed a simulation study.

## Fully simulated datasets
### Genotypes

Note $G$ the genotype matrix with $n$ lines corresponding to individuals (set to 2000) and $m$ columns corresponding to SNPs ($m \in \{8000, 14000, 20000\}$). To code the genotype matrix, we considered an additive genetic model with 0 for a homozygous genotype for the

reference allele, 1 for a heterozygous genotype, and 2 for a homozygous genotype for the alternative allele. Thus, G is a matrix of size $n \times m$ where $G_{ij} \in \{0, 1, 2\}$.

To mimic the linkage disequilibrium (LD) structure, we considered a model adapted from the work of *Wu et al. (2009)*. The full genotype of each individual was generated from an m-dimensional multivariate normal distribution: $G_i^* \sim \mathcal{N}_m(0, \Sigma)$. The correlation matrix $\Sigma$ is block-diagonal with blocks of size $B = 10$ and within each block, all variables are equicorrelated at level $\rho$. The different values considered for $\rho$ are: 0, 0.10, 0.20, 0.35, 0.5 and 0.75.

To obtain the genotypes, these continuous variables were discretized following the Hardy Weinberg equation: $p^2 + q^2 + 2pq = 1$ where, for each SNP, $p$ is the frequency of one of the two possible alleles and $q = 1 - p$.

From here, we arbitrarily set $p$ as the MAF, so that $p \leq q$. The MAF of the $m_0$ non-causal variants were generated from a uniform distribution between 0.01 and 0.5 ($U[0.01, 0.5]$). The $m_1$ causal variants were divided into four distinct subsets in which the MAF were generated from the following distributions:

- Group 1 (Rare SNPs): $U[0.01, 0.05]$
- Group 2 (Medium-Rare SNPs): $U[0.05, 0.15]$
- Group 3 (Medium SNPs): $U[0.15, 0.25]$
- Group 4 (Common SNPs): $U[0.30, 0.40]$

The number of SNPs in each subgroup of causal variants was obtained by the Euclidean division quotient of $m_1$ by 4 with $m_1 \in \{5, 10, 15, 20, 25, 50, 100, 150\}$. The remainder of this division was added to group 4. The effect size simulation is described in the next section.

In the end, we set for each SNP:

- $G_{ij} = 2$ if $G_{ij}^* < q_{p^2, N(0,1)}$,
- $G_{ij} = 1$ if $q_{p^2} < G_{ij}^* < q_{(1-p)^2, N(0,1)}$,
- $G_{ij} = 0$ if $q_{(1-p)^2, N(0,1)} < G_{ij}^*$,

where $q_{., N(0,1)}$ is the quantile function of the standard normal distribution.

### Phenotypes

Let $Y = (y_1, \ldots, y_n)$ be the studied phenotype. In our simulation setting, we considered quantitative and binary variables corresponding to studies for quantitative trait and case-control design, respectively.

*Quantitative trait design.* The values of the quantitative trait were generated from a linear regression model. Thus, for each individual $i$:

$$Y_i = \sum_{j=1}^{m} G_{ij} \beta_j + \epsilon_i \text{ for } j = (1, \ldots, m) \tag{4}$$

where the residuals $\epsilon_i$ were generated from a normal distribution $\mathcal{N}(0, \sigma^2)$. To calibrate the strength of the association, $\sigma^2$ was set as a function of the coefficient of determination

**Table 2  Values of effect sizes ($\beta$) of SNPs for quantitative and binary traits into three scenarios.**

|  |  | Non causal SNP | Rare causal SNP | Medium-Rare causal SNP | Medium causal SNP | Common causal SNP |
|---|---|---|---|---|---|---|
| **Quantitative Trait** | Scenario 1 | 0 | 4 | 3 | 2 | 1 |
|  | Scenario 2 | 0 | 1 | 2 | 3 | 4 |
|  | Scenario 3 | 0 | 2 | 2 | 2 | 2 |
| **Binary Trait** | Scenario 1 | 0 | log(2.2) | log(1.8) | log(1.5) | log(1.3) |
|  | Scenario 2 | 0 | log(1.3) | log(1.5) | log(1.8) | log(2.2) |
|  | Scenario 3 | 0 | log(1.5) | log(1.5) | log(1.5) | log(1.5) |

$R^2$ as in *Stanislas, Dalmasso & Ambroise (2017)*:

$$\sigma_i^2 = \frac{(R^2 - 1)\sum(G_{ij}\beta_j - \bar{Y}_i)^2}{R^2(2 - n)} \tag{5}$$

We set $R^2$ values to 0.2.

*Case-control design.*  For the case-control design, phenotypes were generated from a logistic regression model. Thus, for each individual $i$:

$$\mathbb{P}(Y_i = 1|G_{ij}) = \frac{e^{\beta_0 + G_{ij}\beta_j}}{1 + e^{\beta_0 + G_{ij}\beta_j}} \text{ for } j = (1,\ldots,m) \tag{6}$$

where $\beta_0$ is the intercept corresponding to the expected mean value of $Y$ when all $G = 0$. We set $\beta_0$ so as to obtain balanced proportions of cases and controls in the samples.

*Effect size.*  For both settings (quantitative and binary), the coefficients $\beta_j$ correspond to the effect size of SNP $j$ on the phenotype. For non-causal markers, we set $\beta_j = 0$. For causal variants, we considered three different scenarios (as described in Table 2). The reference scenario (scenario 1) represents the motivating context in which rare causal variants have a greater effect than common variants. For a fair evaluation of the different methods, we also considered two other scenarios: scenario 2 in which common variants have a greater effect than less frequent variants, and scenario 3 in which all $\beta_j$ are equal.

### Larger numbers of tested hypotheses

To evaluate the different procedures with more realistic numbers of hypotheses in GWAS, we also simulated datasets with larger $m$ and $m_1$ values ($m \in \{100000, 200000, 500000\}$ and $m_1 \in \{100, 150, 250\}$) with $R^2 = 0.5$. However, due to excessive computational time for generating the data, we only considered the independent case with quantitative traits.

### Simulation based on real dataset

To mimic a more realistic correlation structure, we also simulated data based on a real dataset from a study on HIV infection (*Dalmasso et al., 2008*). In this study, 307,851 SNPs measured for 605 individuals were analyzed in order to identify new genetic variants associated with plasma HIV-RNA and cellular HIV-DNA levels.

For the reasons of computation time mentioned above, we restricted our simulation to the genotype matrix corresponding to chromosome 6, which has been widely reported in
the literature. All SNPs with more than 10 missing values were removed and the remaining missing values were imputed using the k-nearest neighbors method with $k = 1$ and the Euclidean distance.

From the imputed genotype matrix, MAF were calculated and SNPs were divided into four groups according to their MAF values: the first corresponding to the 558 SNPs with MAF between 0.01 and 0.05, the second corresponding to the 4,909 SNPs with MAF between 0.05 and 0.15, the third corresponding to the 6,674 SNP with MAF between 0.15 and 0.30 and the fourth corresponding to the 7,840 SNPs with MAF larger than 0.30.

Causal variants were randomly drawn in each group in the same proportions as for the fully simulated datasets. To set their $\beta$ values (effect sizes), we first estimated the regression coefficient of all significant SNPs when applying the wBH method to the original dataset at level $\alpha = 0.05$. Then, we considered the absolute values of the four quartiles of the empirical distribution of these estimated coefficients to define the effect size of causal variants in each of the four groups. As for the fully simulated datasets, we considered scenario 1 in which rare causal variants have a greater effect than common variants, scenario 2 in which common variants have a greater effect than less frequent variants, and scenario 3 in which all effects are equally distributed among the four groups. Finally, the phenotypes were generated from a linear regression model with $R^2 = 0.8$.

## Covariates and package versions

Once the phenotypes and genotypes were generated, we applied the multiple testing procedures described in the previous section at a nominal *FDR* level of 5%. For each configuration, we simulated 500 datasets.

### Weights and covariates

In this work, we sought to prioritize the detection of rare variants having strong genetic effects in GWAS. Thus, for wBH, we set the weights analogously to wBHa by considering $w_i = \frac{m}{\sum_{j=1}^{m} \frac{1}{x_j}} \times \frac{1}{x_i}$ (with $x_i$ being the MAF). For the other weighting procedures, the MAF was used as the informative covariate.

However, as previously mentioned, other informative covariates may be considered. For example, we illustrate the use of the proposed method to prioritize common variants by replacing MAF by 1/MAF for all methods (except for CAMT for which we used $\log(1/MAF)$ to avoid computational problems in the EM-algorithm due to large values of the covariate). In addition, to evaluate whether the proposed method is robust when the covariate is completely uninformative, we applied all procedures with a covariate drawn from a uniform distribution between 0 and 1 ($U[0,1]$).

### Package versions

Packages, their versions, and functions used for all analyses are displayed in Table 3.

## Evaluation criteria
### Overall power

To evaluate the ability of the procedures to detect true associations, for each configuration, we estimated the average power $\mathbb{E}\left(\frac{TP}{m_1}\right)$ by the empirical mean (and its corresponding

**Table 3 Procedures compared.**

| Procedure | R package | Function | Version | Reference |
|-----------|-----------|----------|---------|-----------|
| BH | stats | p.adjust | 4.2.1 | *Benjamini & Hochberg (1995)* |
| qvalue | qvalue | qvalue | 2.28.0 | *Storey & Tibshirani (2003)* |
| FDRreg | FDRreg | FDRreg | 0.2.1 | *Scott et al. (2015)* |
| swfdr | swfdr | lm_qvalue | 1.22.0 | *Boca & Leek (2018)* |
| IHW | ihw | ihw | 1.24.0 | *Ignatiadis et al. (2016)* |
| CAMT | CAMT | camt.fdr | 1.1 | *Zhang & Chen (2020)* |

standard error) of the numbers of true discoveries over the 500 simulated datasets divided by $m_1$.

However, for correlated statistics, the definitions of true and false positives are ambiguous since one single genetic marker influencing the phenotype may lead to multiple significant results for correlated markers (*Benjamini, Krieger & Yekutieli, 2006*; *Siegmund, Yakir & Zhang, 2011*; *Brzyski et al., 2017*). To handle this problem, we estimated the power (and the FDR) in correlated datasets by considering clusters of correlated SNPs as units of interest. The clusters were defined according to an estimated correlation coefficient threshold of 0.8.

### Power in subgroups

To evaluate the performance of the methods specifically in the subgroup of rare variants, we also estimated the average power in each subgroup $\mathbb{E}\left(\frac{TP_g}{m_{1_g}}\right), g = (1, 2, 3, 4)$ (where $g = 1$ corresponds to the subgroup of rare causal variants) by the empirical mean (and its corresponding standard error) of the numbers of true discoveries over the 500 simulated datasets divided by the number of causal variants contained in each subgroup.

### FDR control

To assess the FDR control of each procedure, we estimated the FDR by the empirical mean (and its corresponding standard error) of the observed false discovery proportion over the 500 simulated datasets.

## RESULTS

### Simulation results

*Overall power*

Figure 1 shows the overall power for scenario 1 (reference scenario) with independent markers. Results for scenarios 2 and 3 and for correlated markers are available in supplementary materials. As expected, for all procedures, the overall power tends to decrease with the total number of tested hypotheses $m$. It also decreases with $m_1$, since the global effect is distributed among a larger number of SNPs, making the individual effect of each causal variant more difficult to identify. In addition, for correlated datasets, the power tends to increase with $\rho$ in all configurations.

In scenario 1 with independent markers, for small and intermediate values of $m_1$ ($m_1 \leq 25$ and $m_1 \leq 50$ for quantitative and binary phenotypes, respectively), wBHa and

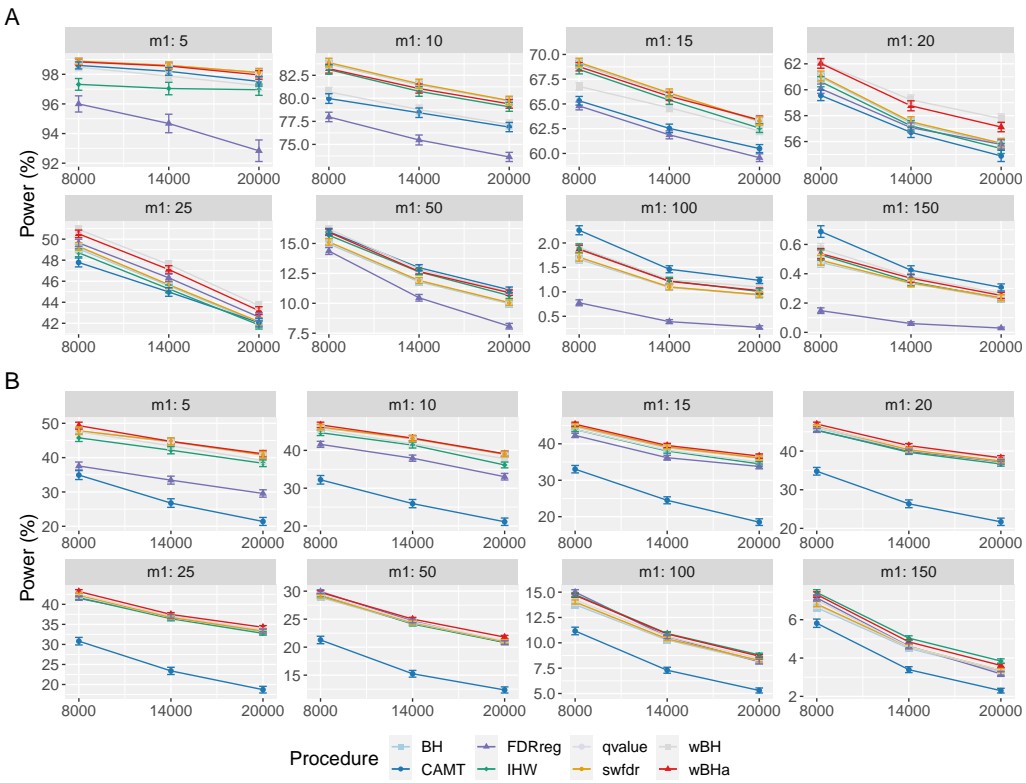

**Figure 1** **Overall power comparison in scenario 1, with independent markers ($\rho = 0$), for different $m$ and $m_1$ values.** Panels A and B display results for quantitative and binary traits, respectively. Vertical bars illustrate standard errors.

wBH tend to be the most powerful procedures, although for the smallest $m_1$ values in the quantitative case ($m_1 \leq 15$), BH, qvalue and swfdr are slightly more powerful (Fig. 1). For larger values of $m_1$, CAMT is the most powerful procedure for quantitative phenotypes (for $m_1 \geq 50$) but the least powerful for binary traits (for $m_1 \geq 100$), the most powerful being IHW. Conversely, FDRreg has good overall power for binary phenotypes but is the least powerful procedure for quantitative phenotypes. Note that while wBHa is not always the most powerful procedure, it has quite good overall power in all configurations in comparison to the other procedures.

In scenarios 2 and 3 with independent markers, for small and intermediate values of $m_1$ ($m_1 \leq 50$), wBHa, wBH, BH, qvalue and swfdr tend to be the most powerful procedures with similar results (except for binary phenotypes in scenario 3 where BH, qvalue and swfdr are more powerful for $m_1 \leq 50$) (Figs. S1 and S2). For larger $m_1$ values ($m_1 \geq 100$), IHW tends to be the most powerful procedure. As in scenario 1, CAMT performs well with quantitative phenotypes but is the least powerful procedure for binary phenotypes.

For correlated markers (Figs. S3, S4 and S5), wBHa is among the most powerful procedures when the value of $m_1$ is intermediate (in scenario 1) or small (in scenarios 2
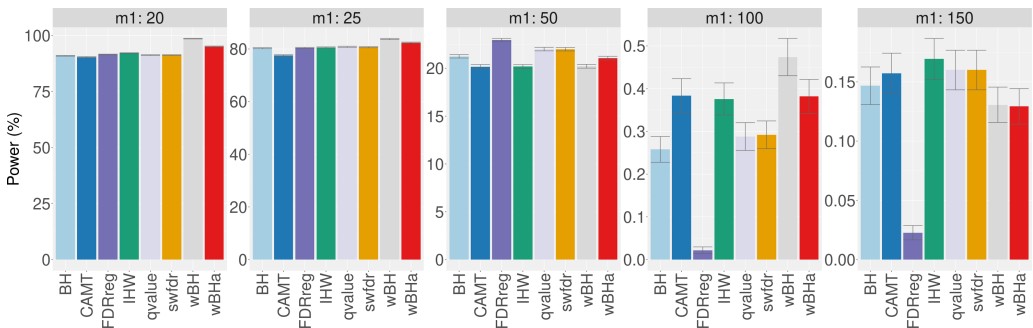

**Figure 2** Overall power comparison in scenario 1, with simulations based on real data, for different $m_1$ values. Vertical bars illustrate standard errors.

and 3). Note that in scenario 1 with binary traits, wBHa tends to be the most powerful procedure in all configurations (Fig. S3).

The results obtained with simulations based on a real dataset are similar to those obtained with fully simulated data (Fig. 2 and Figs. S6 and S7). In all scenarios, for intermediate and small values of $m_1$ ($m_1 \leq 25$), wBHa and wBH tend to be the most powerful procedures while CAMT and IHW are more powerful for large values of $m_1$ ($m_1 \geq 100$).

### Power in subgroups

Figure 3 shows the power of the different procedures to detect associations in the subgroup of rare variants for scenario 1 (reference scenario) with independent markers. Results for scenarios 2 and 3 and for correlated markers are available in supplementary materials. As for the overall power, the power in the subgroup of rare variants tends to decrease with the total number of tested hypotheses $m$ and with the number of causal SNPs $m_1$ for each configuration. In addition, for correlated markers, the power tends to increase with the $\rho$ value in all configurations.

In scenario 1 with independent markers (Fig. 3), the wBH procedure, which is designed particularly for this context, is the most powerful procedure in almost all settings. However, our procedure wBHa shows quite large power compared to the other procedures. For large values of $m_1$ ($m_1 \geq 50$), CAMT tends to be the most powerful procedure for quantitative phenotypes, but it is the least powerful one for binary phenotypes. Conversely, FDRreg performs well for binary phenotypes but it is the least powerful for quantitative phenotypes.

Interestingly, in intermediate scenarios (scenarios 2 and 3), our procedure and wBH tend to be the most powerful in the subgroup of rare variants for all configurations (Figs. S8 and S9). However, as expected, the powers of all procedures for detecting associations of rare variants are low in scenario 2, where the smallest effects are attributed to rare variants.

In the correlation case, for all scenarios we obtained results similar to those obtained in the independent case (Figs. S10, S11 and S12). Thus, wBHa and wBH tend to be the most powerful procedures in the subgroup of rare variants for all settings, except for $m_1 \geq 50$ for quantitative traits in scenario 1.

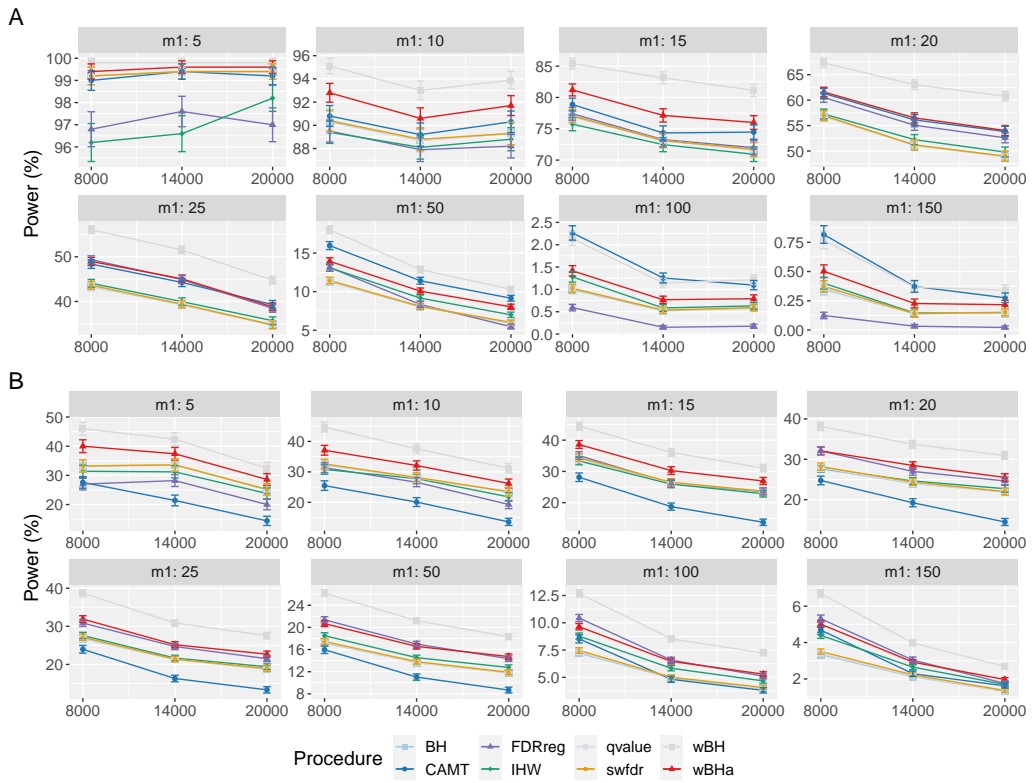

**Figure 3** **Power comparison in subgroup of rare variants in scenario 1 with independent markers ( $\rho = 0$ ), for different $m$ and $m_1$ values.** Panels A and B display results for quantitative and binary traits, respectively. Vertical bars illustrate standard errors.

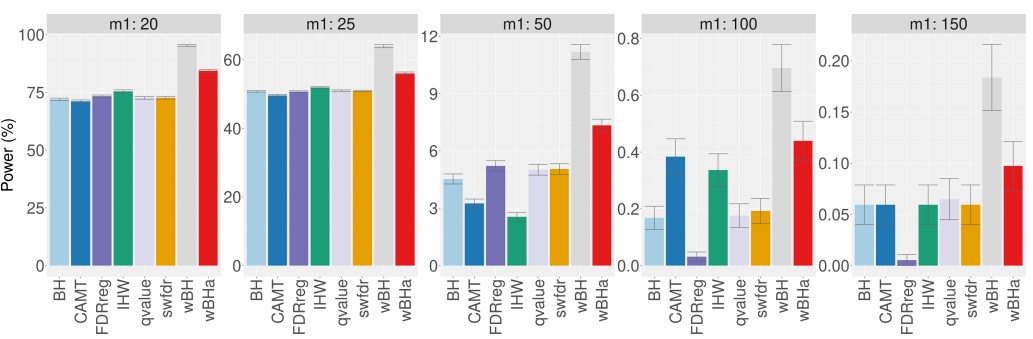

**Figure 4** **Power comparison in subgroup of rare variants in scenario 1, with simulations based on real data, for different $m_1$ values.** Vertical bars illustrate standard errors.

When considering simulations based on a real dataset (Fig. 4 and Figs. S13 and S14), wBHa is among the most powerful procedures in all settings, the most powerful being wBH.
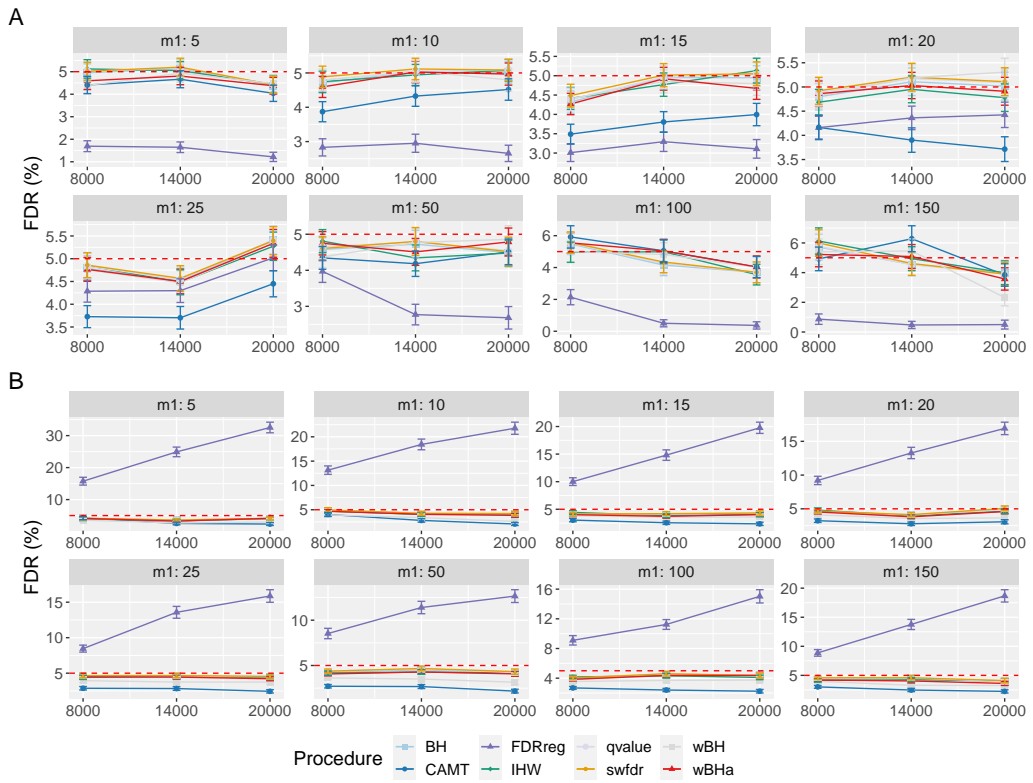

**Figure 5  FDR comparison in scenario 1, with independent markers ( $\rho = 0$), for different $m$ and $m_1$ values.** Panels A and B display results for quantitative and binary traits, respectively. Red dashed line corresponds to target FDR level (5%). Vertical bars illustrate standard errors.

### FDR control

Figure 5 displays the estimated FDR for all procedures in scenario 1 with independent markers. These results indicate a good control of the FDR for all procedures in all settings (except for FDRreg with binary phenotypes). Indeed, for all procedures except FDRreg, the estimated FDR is lower than 0.05 or slightly larger than this threshold. This can be explained by the fact that in our simulated settings, the numbers of rejections tend to be small, leading to a quite large variability of the false discovery proportion. Thus, even with the BH procedure (for which the FDR control has been theoretically proven for independent tests), the estimated FDR is slightly larger than the threshold 0.05 in some settings. Similar results were obtained for scenarios 2 and 3 (Figs. S15 and S16).

In the case with correlations between variants (Fig. 6), the estimated FDR increases with $\rho$ in all configurations. Similar results were obtained in scenarios 2 and 3 (Figs. S17 and S18). In addition, the estimated FDR obtained with simulations based on a real dataset (Figs. S19, S20 and S21) tend to be large for all procedures. These results illustrate the difficulty to define and to control the FDR when tested hypotheses are correlated. Nevertheless, as the estimated FDR are similar from one procedure to another, the power comparisons remain relevant.

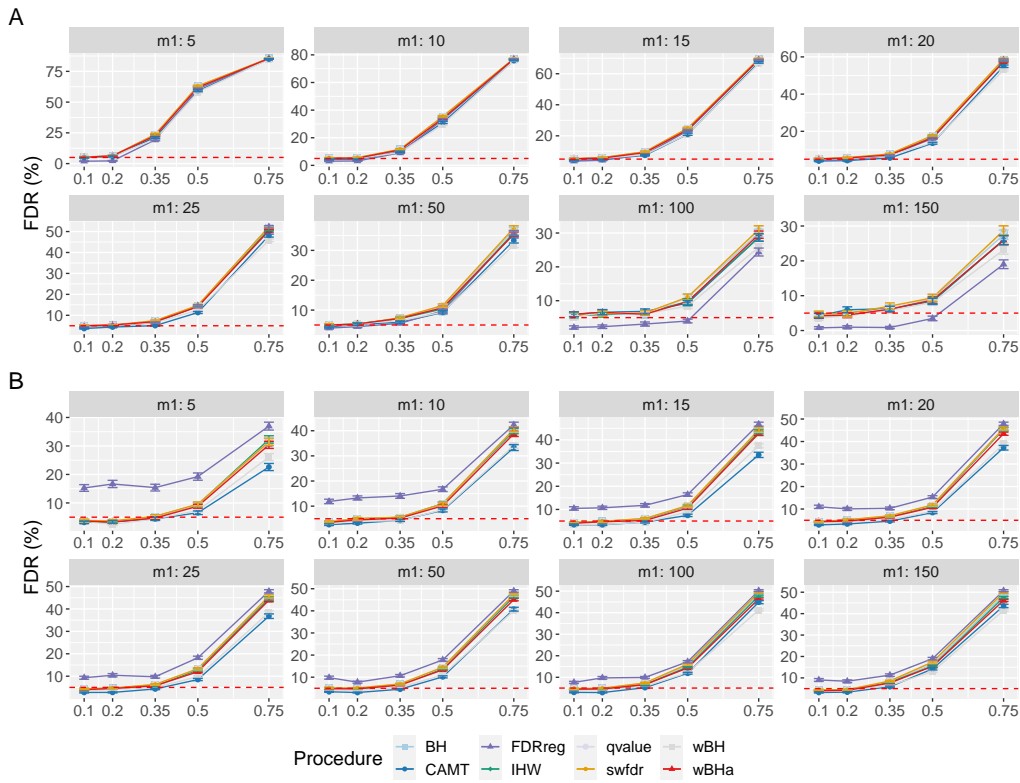

**Figure 6** **FDR comparison in scenario 1, with correlated markers, for different $\rho$ and $m_1$ values with $m = 8000$.** Panels A and B display results for quantitative and binary traits, respectively. Red dashed line corresponds to target FDR level (5%). Vertical bars illustrate standard errors.

### Larger numbers of tested hypotheses

When considering larger values of $m$ and $m_1$ for quantitative phenotypes, wBHa belongs to the three most powerful procedures in scenario 1 while maintaining good overall power in scenarios 2 and 3 (Fig. S22). wBHa is also one of the three most powerful procedures in the subgroup of rare variants in scenario 1, while in scenarios 2 and 3, CAMT, IHW and swfdr tend to be slightly more powerful, particularly for large values of $m_1$ (Fig. S23). Figure S24 indicates a quite good control of the FDR for all procedures.

### Other covariate

When MAF is replaced by 1/MAF to prioritize common variants, wBHa is one of the most powerful procedures for detecting common variants in all scenarios (Figs. S25, S26 and S27) while maintaining an overall power similar to that of the unweighted procedure BH (Figs. S28, S29 and S30). The FDR is controlled by all procedures except FDRreg for binary phenotypes (Figs. S31, S32 and S33). Note that the use of $log(1/MAF)$ in CAMT makes it possible to avoid computational problems but leads to very similar results in the cases where CAMT worked with 1/MAF (data not shown).

When using an uninformative covariate, the different procedures remain valid since the FDR is controlled at the desired level (Figs. S34, S35 and S36). All weighted procedures

tend to have a lower overall power than the unweighted BH procedure, although wBHa is the one with the smallest loss (Figs. S37, S38 and S39).

### Real dataset analysis

To illustrate the results obtained with the simulated data, we applied the different procedures on a publicly available dataset from the study conducted by (*Liu et al., 2017*). One of the objectives of that study was to identify new potential genetic variants that influence Crohn's disease. The genotypes of 659,636 SNPs for 98 individuals together with the Paneth cell phenotype are available in the Gene Expression Omnibus (GEO) database (GSE90102).

For our analysis, we only considered autosomal chromosomes. Therefore, standard quality controls were applied: we removed all SNPs with a call rate of less than 95%, all SNPs with a significant deviation from the Hardy-Weinberg equilibrium (pvalue less than $10^{-5}$), and all SNPs with a MAF less than 0.01. The MAF distribution of all SNPs is presented in Fig. S40. After applying these filters, 607,720 SNPs were analyzed for GWAS. To test the association between the genotypes and the percentage of abnormal Paneth cells (quantitative phenotype), we used a classical linear regression model. Then the different weighted multiple testing procedures were applied.

Figure 7 shows the total number of rejections for each procedure for different categories of MAF. While CAMT and IHW, which tend to be the most powerful procedures in our simulation study for quantitative phenotypes, led to the largest total numbers of rejected null hypotheses ($R = 111$ and $R = 109$, respectively), wBHa identified 106 significant results while the procedures leading to the smallest number of rejections were BH, qvalue and swfdr, which identified the same 43 markers. In addition, wBHa produced the largest number of rejections for SNPs with a MAF lower than 0.02.

Note that while wBHa is not the most powerful procedure, it identified six specific SNPs (Fig. 8) that could not be selected by the other procedures. Interestingly, two of these SNPs, rs3772479 and rs2270569, are located in the FHIT and KIF9 genes, respectively, which have been reported to play an important role in inflammatory bowel disease (IBD) (Crohn's disease being a type of IBD) (*Skopelitou et al., 2003*; *Xu & Qiao, 2006*; *Wierzbicki et al., 2009*; *Wang et al., 2018*).

## DISCUSSION

In this study, we evaluated recent weighted multiple testing procedures in the context of genome-wide association studies. We also introduced a new procedure called wBHa that aims to prioritize the detection of genetic markers with a low MAF while letting the procedure adapt a weighting function in order to maximize the overall power.

For independent datasets, wBHa performed well in the simulation study compared to the other procedures with a quite good overall power in all simulated configurations. As noted by *Korthauer et al. (2019)*, we found that IHW and CAMT perform better when the proportion of non-null hypotheses increases. However, the proportion of non-null hypotheses is hard to estimate and using these two procedures in a context where only few markers are associated to the phenotype may lead to a reduction in overall power. The

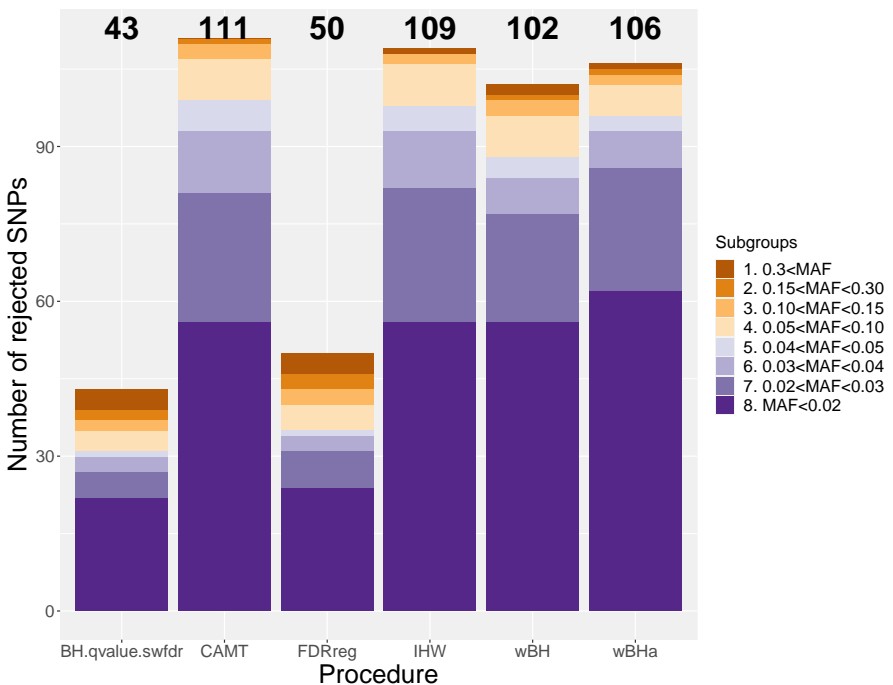

**Figure 7  Number of rejected SNPs for subgroups of SNPs for each procedure.**

fact that wBHa tends to be the most powerful procedure for smaller proportions of causal variants in scenarios 2 and 3 than in scenario 1 can be explained by the increasing difficulty to detect causal variants when their effect size is smaller.

When considering the power to detect associations within the subgroup of rare variants, the non-optimal wBH procedure is the most powerful in all scenarios. However, wBHa is not only powerful in the subgroup of rare variants but also has good overall power. These results demonstrate the value of the optimization parameter *a* in the wBHa procedure.

Concerning FDR control, most procedures seem to correctly control the error criterion for independent datasets, although in some cases, the estimated FDR for all procedures (including BH) is slightly larger than the threshold. This can be explained by the small number of rejections, which leads to a large variability of the false discovery proportion. However, FDRreg does not appear to control the FDR in the case-control design, which is consistent with similar results obtained by *Boca & Leek (2018)*; *Korthauer et al. (2019)* and *Zhang & Chen (2020)* for some configurations.

For correlated datasets (fully simulated or based on a real dataset), we obtained similar results in terms of power. Thus, wBHa showed good performance compared to the other procedures. As expected, a loss of FDR control is observed with all procedures when the correlations increase, and the difficulty of defining true and false positives remains a challenge. However, in practice, the influence of correlations on FDR control may be limited by using methods such as LD pruning (*Purcell et al., 2007*).

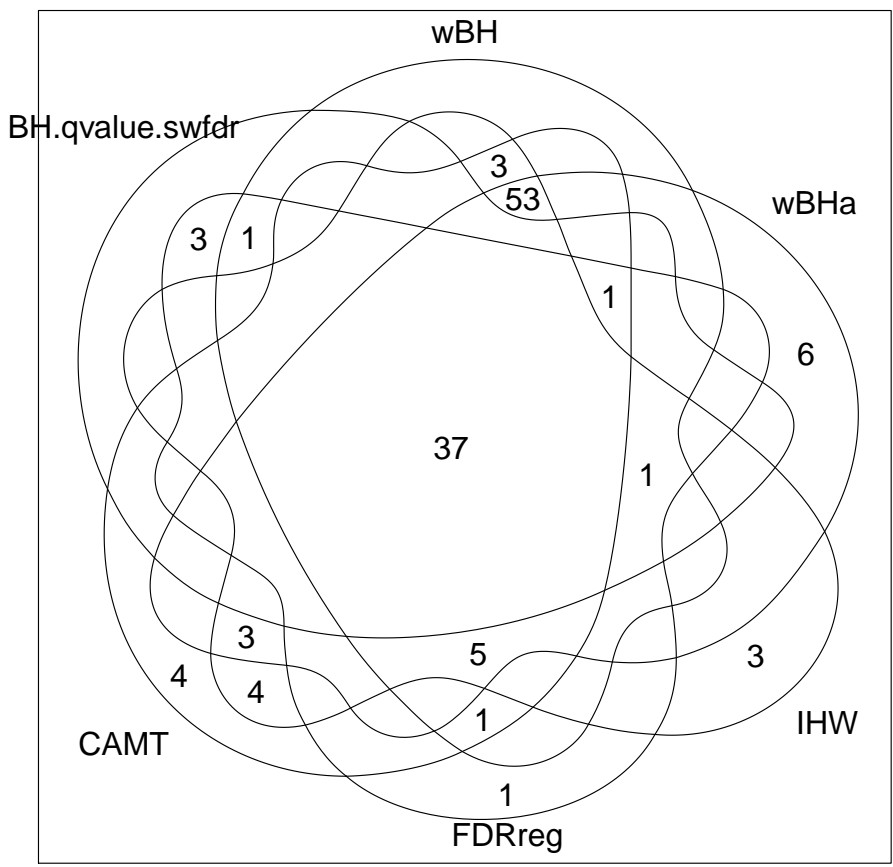

**Figure 8    Venn diagram of selected SNPs for all procedures.**

To sum up, although wBHa is not the most powerful in all configurations, it showed good performance compared to other procedures in all configurations, not only in terms of overall power but also regarding the power to detect associations in the subgroup of rare variants. In particular, in scenarios 2 and 3 for which rare variants have moderate or low effect sizes, wBHa proved to be powerful in the subgroup of rare SNPs, thereby testifying to its value. Thus, wBHa allows the detection of rare variants while having overall power similar to that of that of other procedures, whatever the size effect of rare variants.

To illustrate the results obtained with simulated data, we applied the different procedures to a real public dataset on Crohn's disease. The results were consistent with those obtained with simulated data. The weighted procedures had good performance in terms of power, particularly for procedures based on informative covariates. Moreover, wBHa identified six specific rare variants that could not be selected by the other procedures. Among them, two markers are located in genes FHIT and KIF9, which have been reported to be involved in IBD, suggesting that they could be true associations. These results underline the value of wBHa in real data applications.

In conclusion, adaptive weighted multiple testing procedures based on informative covariates show great promise in the context of genome-wide association studies. Our new

procedure wBHa, which showed good performance in all settings, appears to be a good choice for prioritizing rare variants without loss of overall power.

## AVAILABILITY

The wBHa procedure is implemented in the R package wBHa which is available at https://github.com/obryludivine/wBHa. A second GitHub repository is also available at https://github.com/obryludivine/wBHa_simulation. It contains the programs used to create the simulated datasets and allows our results to be reproduced. These projects have been archived on Zenodo on https://zenodo.org/badge/latestdoi/409590338 and https://zenodo.org/badge/latestdoi/402729574 respectively.

### Funding
This work was supported by the doctorate program of the University of Paris Saclay, France. The funders had no role in study design, data collection and analysis, decision to publish, or preparation of the manuscript.

### Competing Interests
The authors declare there are no competing interests.

### Author Contributions
- Ludivine Obry conceived and designed the experiments, performed the experiments, analyzed the data, prepared figures and/or tables, authored or reviewed drafts of the article, and approved the final draft.
- Cyril Dalmasso conceived and designed the experiments, authored or reviewed drafts of the article, and approved the final draft.

### Data Availability
The wBHa procedure is implemented in the R package wBHa which is available at GitHub:

- Available at https://github.com/obryludivine/wBHa.

- Available at https://github.com/obryludivine/wBHa_simulation. This contains the programs used to create the simulated datasets and allows our results to be reproduced.

These projects are available at Zenodo:

- Ludivine Obry. (2023). obryludivine/wBHa: v0.0.0.9 (v0.0.0.9). Zenodo. Available at https://doi.org/10.5281/zenodo.7702448

- Ludivine Obry. (2023). obryludivine/wBHa_simulation: v0.0.0.9 (v0.0.0.9). Zenodo. Available at https://doi.org/10.5281/zenodo.7702457.

### Supplemental Information
Supplemental information for this article can be found online at http://dx.doi.org/10.7717/peerj.15369#supplemental-information.

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
