# Peer review of "Weighted multiple testing procedures in genome-wide association studies"

_PeerJ, doi:10.7717/peerj.15369_

## Round 0.1 · original submission · Major Revisions

Comments by the reviewers have been received. Rectification of inconsistency with the literature is required. They have also asked for clarity on some points, particularly definition of V and on some statistical parameters.

·

Basic reporting

1. The informative covariate in this work is defined only on its basis of independence under the null hypothesis; this is inconsistent with previous cited work which references additional criteria for informativeness (e.g. as stated by Ignatiadis 2016 as referenced in line 126, an informative covariate is also associated with each test’s power or prior probability of null hypothesis).

2. The nuances of multiple testing correction in the broader context of the GWAS literature are not fully described in the background. In classical GWAS studies, Bonferroni is the most widely used approach. . In addition, there are implications for FDR control in the local correlation structure of genetic variants (linkage disequilibrium).

3. Various spelling typos throughout, and could benefit from moderate editing for sentence style and flow.

Experimental design

1. The basic premise of the weight assignment in wBHa is unclear in the general case. Are the weights in the wBHa also considered ‘informative covariates’? That is, is it assumed that they are independent of the p-values under the null, and/or that they are informative of power? If so, what happens when either assumption is violated?

2. How would one choose these weights in general, aside from the GWAS rare variant context?

3. If using the wBHa procedure for a covariate other than MAF, is the same weight function w(x,a) required to generalize results? What can the authors assert if the weight function were used to prioritize common variants?

4. The range of numbers of hypotheses m considered are quite small. To be realistic, these simulations should also consider much larger numbers.

5. Why does the real data-based simulation only corresponded to full simulation scenario 1? It does not necessarily follow that data-based simulations corresponding to scenarios 2 and 3 would reflect similar findings to the full simulation.

Validity of the findings

1. Since simulations are summarized over multiple replications, and since the authors mention quite large variability in results (line 373), figures should include error bars to indicate the variability in the average results. Without these, it is not possible to interpret the relative ordering of the methods.

2. What weights are used for the wBH procedure in this study? I don’t see this defined anywhere. Without discussion of this choice, justification versus alternatives, and its implications/assumptions, it’s not possible to assess the validity of the provided comparison with the wBHa procedure. In particular, as noted by a study cited in this submission on line 72 (Genovese et al. 2006) the choice of weights is critical to performance of the wBH procedure.

3. In order to claim that the 6 SNPs that only wBHa identified in the real data application are potentially useful, the authors should examine the plausibility of these associations being true positives using prior biological knowledge surrounding these specific variants in this application.

4. How do you explain why power in simulation scenario 1 (Figures 1 and 2) decreases dramatically from the low m1 values (power near 100%) to high m1 values (power less than 1%)? Are similar numbers of rejections made for each of the m1 variations?

Additional comments

1. The increase in FDR with increased correlation is not surprising, and one might wonder if some standard GWAS practices such as LD pruning, would alleviate it.

2. Line 110: V is not defined.

Reviewer 2 ·

Basic reporting

Obry et al introduced a new method to correct for multiple testing for Genome-wide association studies. They tested their new approach with a simulated and real dataset. The simulation study consisted of three different scenarios with 2000 individuals. The simulation study used quantitative and binary traits to compare the power % between the different existing methods. The overall performance of the suggested method indicated that the method is among the best by the authors. In addition, with the real-life dataset, they tested and proved their findings. Overall, the paper was well-done and the topic is of interest to the scientific community. However, the treatment of “rare” SNPs was inconsistent between the simulations and real data analysis – this needs to be addressed before publication.

Experimental design

- Line 244: For the simulation study, the authors define “rare SNPs” as those with MAF between 0.05 and 0.15. Few would consider MAF=0.05 to be rare, especially with a sample size of 2000 individuals. For the real data analysis, SNPs with MAF as low as 0.01 were analyzed (line 392). Could the authors adjust the MAF class to start at 0.01 or below for the first group? Or better, could a new “rare” group for the MAF between 0.01 (or lower) and 0.05 be defined?

Validity of the findings

- Line 343: For figure two, the author compared the wBHa to other procedures (CAMT, IHW, etc.) when m1<25. However, there is no explanation for the m1>25 in lines 342, 343, or 344. Especially for the m1:100, why is the power of wBH and wBHa among the least robust method?

Additional comments

Line 33, 34, 35: Long sentence, not easy to follow. Could you write the sentence shorter and clearer?
Line 110, 111: What is the definition of V? I think V stands for true negatives (TN). Could the authors explain What V stands for?
Line 123: Minor allele frequency can be written as MAF – it was defined earlier. Throughout the manuscript, the authors inconsistently switch between “MAF” and “minor allele frequency”.
Line 400, 401: Could the author illustrate allele frequency distribution in supplementary figures?
Line 442, 443: “wBHa identifies specific rare variants” compared to which procedure? According to figure 7, wBHa rejected more SNPs than all other procedures (MAF<0.02), and for the total number of rejections, wBHa rejected more SNPs than wBH, BH.qvalue.swfdr, FDRreg. Could the author specify the sentence?

The text size for all figures is small, and it is not easy to read the numbers without zooming in. Could the author increase the text size for all figures (Including the supplementary figures) except for 7?

---

## Round 0.2 · accepted · Accept

In light of comments received, the authors have done a satisfactory job. Therefore, I recommend acceptance.

·

Basic reporting

The authors have satisfactorily addressed my concerns in the revised version.

Experimental design

The authors have satisfactorily addressed my concerns in the revised version.

Validity of the findings

The authors have satisfactorily addressed my concerns in the revised version.

Additional comments

The authors have satisfactorily addressed my concerns in the revised version.